# SARS-CoV-2 Seroprevalence among Healthcare Workers in General Hospitals and Clinics in Japan

**DOI:** 10.3390/ijerph18073786

**Published:** 2021-04-05

**Authors:** Tatsuya Yoshihara, Kazuya Ito, Masayoshi Zaitsu, Eunhee Chung, Izumi Aoyagi, Yoshikazu Kaji, Tomomi Tsuru, Takuma Yonemura, Koji Yamaguchi, Shinichi Nakayama, Yosuke Tanaka, Nobuo Yurino, Hideki Koyanagi, Shunji Matsuki, Ryuji Urae, Shin Irie

**Affiliations:** 1Clinical Research Center, SOUSEIKAI Fukuoka Mirai Hospital, Kashiiteriha 3-5-1, Higashi-ku, Fukuoka 813-0017, Japan; shunji-matsuki@lta-med.com; 2SOUSEIKAI Clinical Epidemiological Research Center, Kashiiteriha 3-5-1, Higashi-ku, Fukuoka 813-0017, Japan; kazuya-ito@lta-med.com; 3College of Healthcare Management, Takayanagi 960-4, Setaka-machi, Miyama 835-0018, Japan; 4Department of Public Health, Dokkyo Medical University School of Medicine, 880 Kitakobayashi, Mibu-machi, Shimotsuga-gun, Tochigi 321-0293, Japan; m-zaitsu@dokkyomed.ac.jp; 5SOUSEIKAI Global Clinical Research Center, Kashiiteriha 3-5-1, Higashi-ku, Fukuoka 813-0017, Japan; eunhee-chung@lta-med.com; 6Department of Internal Medicine, SOUSEIKAI Fukuoka Mirai Hospital, Kashiiteriha 3-5-1, Higashi-ku, Fukuoka 813-0017, Japan; i-aoyagi@fukuoka-mirai.jp; 7SOUSEIKAI Hakata Clinic, 6-18 Tenyamachi, Hakata-ku, Fukuoka 812-0025, Japan; yoshikazu-kaji@lta-med.com; 8SOUSEIKAI PS Clinic, 6-18 Tenyamachi, Hakata-ku, Fukuoka 812-0025, Japan; tomomi-tsuru@lta-med.com; 9SOUSEIKAI Sumida Hospital, 1-29-1, Honjo, Sumida-ku, Tokyo 130-0004, Japan; takuma-yonemura@lta-med.com; 10SOUSEIKAI Nishikumamoto Hospital, 1012 Koga, Tomiaimachi, Minami-ku, Kumamoto 861-4157, Japan; kouji-yamaguchi@nishikuma.com; 11SOUSEIKAI Miyata Hospital, 1636 Honjo, Miyawaka, Fukuoka 823-0003, Japan; snakayam@lta-med.com; 12SOUSEIKAI Kanenokuma Hospital, 3-24-16 Kanenokuma, Hakata-ku, Fukuoka 812-0863, Japan; y.tanaka@kanenokuma.jp; 13SOUSEIKAI Shinyoshizuka Hospital, 7-6-29, Yoshizuka, Hakata-ku, Fukuoka 812-0041, Japan; n-yurino@lta-yoshizuka-g.jp; 14SOUSEIKAI Dodo Clinic, 1-31-13, Nakaikegami, Ota-ku, Tokyo 146-0081, Japan; hideki-koyanagi@lta-med.com; 15SOUSEIKAI, 3-5-1 Kashiiteriha, Higashi-ku, Fukuoka 813-0017, Japan; ryuji-urae@lta-med.com (R.U.); shin-irie@lta-med.com (S.I.)

**Keywords:** antibody, COVID-19, healthcare workers, SARS-CoV-2, seroprevalence

## Abstract

Coronavirus disease 2019 (COVID-19) has become a serious public health problem worldwide. In general, healthcare workers are considered to be at higher risk of COVID-19 infection. However, the prevalence of COVID-19 among healthcare workers in Japan is not well characterized. In this study, we aimed to examine the seroprevalence of severe acute respiratory syndrome coronavirus-2 (SARS-CoV-2) antibodies among 2160 healthcare workers in hospitals and clinics that are not designated to treat COVID-19 patients in Japan. The prevalence of SARS-CoV-2 immunoglobulin G was 1.2% in August and October 2020 (during and after the second wave of the pandemic in Japan), which is relatively higher than that in the general population in Japan (0.03–0.91%). Because of the higher risk of COVID-19 infection, healthcare workers should be the top priority for further social support and vaccination against SARS-CoV-2.

## 1. Introduction

Coronavirus disease 2019 (COVID-19) has become a serious public health problem worldwide. In Japan, the confirmed cases of COVID-19 remained low as of December 2020 compared with those in Europe or northern America. The seroprevalence of COVID-19 in Japan’s general populations has been reported to be 0.03–0.40% from June to September 2020 and 0.14–0.91% in December 2020 [1,2]. In general, healthcare workers are considered to be at higher risk of COVID-19 infection [3]. However, two studies have shown a low seroprevalence of COVID-19 (0–0.16%) among healthcare workers engaged in the direct diagnosis, treatment, and care of patients with COVID-19 during the first wave of the epidemic in Japan [4,5]. This low prevalence may reflect effective infection control and high awareness of infection prevention due to the increased infection risk in treating COVID-19 patients [4,5]. However, the prevalence among healthcare workers in hospitals that are not designated to treat COVID-19 patients in Japan is not well characterized [5,6]. Here, we aimed to determine the seroprevalence in August and October 2020 (during and after the second wave of the pandemic in Japan) among workers in general hospitals and clinics in Japan.

## 2. Materials and Methods

### 2.1. Study Design and Sites

A multicenter prospective study was conducted in nine general hospitals and clinics of the SOUSEIKAI Medical Group: Fukuoka Mirai Hospital (FMH), Hakata Clinic (HC), PS Clinic (PC), Sumida Hospital (SH), Miyata Hospital (MH), Kanenokuma Hospital (KH), Shinyoshizuka Hospital (SYH), Nishikumamoto Hospital (NH), and Dodo Clinic (DC). These hospitals/clinics are not designated to treat COVID-19 patients. FMH Clinical Research Center, HC, NH Clinical Pharmacology Center, and SH are specialized facilities for clinical trials (mainly Phase 1 clinical trials). FMH, PC, MH, KH, and SYH are located in Fukuoka prefecture; NH is located in Kumamoto prefecture; and DC and SH are located in Tokyo.

### 2.2. Ethics

This study was approved by the SOUSEIKAI Hakata Clinic Institutional Review Board (approval number: N-81) and registered in the UMIN Clinical Trial Registry (registration number: UMIN000041262). The people working in the hospitals/clinics of the SOUSEIKAI Medical Group were invited to participate in the study. All participants provided written informed consent.

### 2.3. Laboratory Assay

Severe acute respiratory syndrome coronavirus-2 (SARS-CoV-2)-specific immunoglobulin (Ig)M and IgG antibodies in the venous blood were assessed using an immunochromatographic assay kit (2019-nCoV Ab Test [Colloidal Gold], INNOVITA Biological Technology Co., Ltd., Tangshan, China) following the manufacturer’s instructions. The assay kit was granted Emergency Use Authorization by the United States Food and Drug Administration. According to the manufacturer’s instructions, the clinical sensitivity and specificity (95% confidence interval) were estimated to be 87.3% (80.4% to 92.0%) and 100% (94.2% to 100%), respectively. Blood samples were collected in August and/or October 2020.

### 2.4. Job Title, COVID-19 Symptoms, and Diagnosis

In each survey, a questionnaire written in Japanese was used to obtain the following data: job title, presence of suspected COVID-19 symptoms since February 2020, and history of COVID-19 diagnosis.

The participants’ occupations were divided into eight categories: nurses (including nurse assistants), physicians, technicians (laboratory technicians, radiology technicians, pharmacists, clinical engineers, dental hygienists, physical therapists, occupational therapists, speech therapists, and acupuncturists), nursing care staff, office workers, receptionists, employees in clinical research units, and others (drivers, security personnel, nursery school teachers, shop workers, sanitary workers, nutritionists, and food service staff).

We defined the presence of suspected COVID-19 symptoms since February 2020 as the participant having had any of the following symptoms: fever, runny nose/stuffy nose, sore throat, cough, sputum, difficulty in breathing, severe fatigue, altered sense of taste and/or smell, erythematous rash on fingers/toes, joint pain/muscle pain, headache, nausea/vomiting, or diarrhea.

History of COVID-19 diagnosis was based on the answer to the question, “Have you been diagnosed with COVID-19? (yes/no).”

### 2.5. Statistical Analysis

Using descriptive statistics, the difference in seropositive rates across background characteristics was assessed using a Chi-square test. Two-tailed *p*-values of <0.05 were considered significant. Statistical analyses were performed using JMP Pro 15 (SAS Institute Inc. Japan, Tokyo, Japan).

## 3. Results

A total of 2160 SOUSEIKAI workers, aged 20–83 years (mean = 41.9, standard deviation = 12.7; women: 1547 [71.6%]), underwent at least one antibody test in August and/or October. IgM and/or IgG SARS-CoV-2 antibodies were detected in 33 (1.5%) and 27 (1.3%) participants in August and October, respectively (Table 1). IgG antibodies against SARS-CoV-2 were detected in 25 (1.2%) and 24 (1.2%) participants in August and October, respectively. After excluding one facility, where nosocomial infections occurred in April 2020, 13 participants (0.8%) demonstrated IgG positivity in August and October (Table 1). Of the eight participants who showed IgM positivity alone in August, none reported IgG positivity in October. Among the participants who showed IgG positivity in August, 95.5% (21/22) demonstrated IgG positivity in October.

The prevalence of IgG antibodies was higher among the 20–29-year and 60–69-year age groups compared with other age groups (*p* = 0.006), whereas no significant difference was observed between women and men (*p* = 0.69) (Table 2). The prevalence of IgG antibodies was relatively, but not significantly, higher among nurses, nursing care staff, and receptionists (*p* = 0.25) (Table 2). Among the 28 participants who had at least one IgG-positive test result, 17 (60.7%) had suspected symptoms of COVID-19 since February 2020. Among the 14 participants diagnosed with COVID-19 prior to the antibody tests, 10 (71.4%) tested positive for IgG antibodies (Table 2). Among the 2146 participants who had not been diagnosed with COVID-19, 18 (0.8%) tested positive for IgG antibodies.

## 4. Discussion

The prevalence of SARS-CoV-2 antibodies varies among countries and cities. Europe and northern America have a high prevalence of SARS-CoV-2 antibodies, whereas Eastern Asia has a relatively low prevalence [7]. In Japan, the seroprevalence of COVID-19 in the general population has been reported to be 0.03–0.40% from June to September 2020 and 0.14–0.91% in December 2020 [1,2]. In this study, the seroprevalence of SAR-CoV-2 IgG antibodies among healthcare workers in hospitals and clinics that were not designated to treat COVID-19 patients was 1.2% in August and October 2020. It has been reported that the prevalence of SARS-CoV-2 antibodies in healthcare workers is higher than that in the general population [3,8]. Our study also demonstrated that the prevalence of SARS-CoV-2 antibodies was relatively higher in healthcare workers than in the general population in Japan.

The 20–29-year and 60–69-year age groups had high prevalence of SARS-CoV-2 IgG antibodies. Among the five participants in the 60–69-year age group who tested positive for IgG antibodies, four worked in the facility where nosocomial infection occurred in April 2020; therefore, the high prevalence among the 60–69-year age group was possibly due to nosocomial infection. The routes of infection in the 20–29-year age group have not been fully elucidated. However, the high occupational and daily activities and the demographic characteristics of asymptomatic carriers in the young age group were considered to be contributing factors.

The level of SARS-CoV-2 antibodies remains high for at least a few months after developing a COVID-19 infection [9,10,11]. A recent study reported high levels of SARS-CoV-2 antibodies (69.0–91.4%) 8 months after asymptomatic or mild SARS-CoV-2 infection [12]. In this study, 95% (21/22) of participants who had IgG antibodies in August tested positive for IgG antibodies in October. Among the participants who had a history of acquiring COVID-19, 71.4% (10/14) showed IgG positivity up to 6 months after infection. These results support those of previous studies, which reported that IgG antibodies can be detected in the majority of patients a few months after SARS-CoV-2 infection.

Based on the results of this study, several infection-protection measures were implemented in our medical group. For example, we conducted SARS-CoV-2 PCR tests in all medical workers once a month for the early detection of infected workers. We also instructed medical workers to conduct basic infection control measures not only in hospitals but also in daily life and to a have high awareness of infection prevention as healthcare workers to protect patients and co-workers. We believe that revealing the current prevalence of SARS-CoV-2 is important for conduct appropriate control measures. After December 2020, Japan experienced the third wave of the pandemic. Therefore, further studies are needed to examine the prevalence of SARS-CoV-2 in Japanese healthcare workers.

One limitation of this study was that only one antibody assay kit was used. As PCR tests or antigen tests were not conducted in participants with SARS-CoV-2 IgM or IgG antibodies, it remained uncertain whether false-positive data were included. In particular, among the eight participants who tested positive for IgM antibodies alone in August, none tested positive for IgG antibodies in October. We recruited individuals working at SOUSEIKAI medical group facilities for this study; of them, 91.6% (2142 out of 2338) in August and 90.0% (2081 out of 2311) in October participated in this study. Thus, the effect of selection bias is limited.

## 5. Conclusions

The prevalence of SARS-CoV-2 IgG antibodies among healthcare workers in August and October 2020 in Japan was 1.2%, which was relatively higher than that in Japan’s general population. Healthcare workers are at higher risk of infection; hence, they should be the top priority for further social support and SARS-CoV-2 vaccination.

## Figures and Tables

**Table 1 ijerph-18-03786-t001:** Prevalence of SARS2-CoV2 antibodies in August or October 2020 stratified by region.

	August 2020				October 2020			
Region (Prefecture)	n	Positive, n (%)			n	Positive, n (%)		
		IgM	IgG	IgM and/or IgG		IgM	IgG	IgM and/or IgG
Fukuoka	1746	10 (0.6)	25 (1.4)	33 (1.9)	1698	3 (0.2)	24 (1.4)	27 (1.6)
Kumamoto	321	0 (0)	0 (0)	0 (0)	308	0 (0)	0 (0)	0 (0)
Tokyo	75	0 (0)	0 (0)	0 (0)	75	0 (0)	0 (0)	0 (0)
Total	2142	10 (0.5)	25 (1.2)	33 (1.5)	2081	3 (0.1)	24 (1.2)	27 (1.3)
Nosocomial infection (−)	1654	10 (0.6)	13 (0.8)	21 (1.3)	1595	3 (0.2)	13 (0.8)	16 (1.0)
Nosocomial infection (+)	488	0 (0)	12 (2.5)	12 (2.5)	486	0 (0)	11 (2.3)	11 (2.3)
*p*-value ^a,b^		0.09	0.003	0.06		0.34	0.009	0.03

^a^ Pearson’s Chi-square test; ^b^ Comparison between the facilities where nosocomial infection occurred and those where nosocomial infection did not occur.

**Table 2 ijerph-18-03786-t002:** Prevalence of SARS-CoV-2 antibodies in August and/or October 2020 stratified by COVID-19 history, decade age group, sex, and job categories.

Characteristics	n	IgM Positive n (%)	IgG Positive n (%)	IgM and/or IgG Positive, n (%)
Total	2160	10 (0.5)	28 (1.3)	36 (1.7)
COVID-19 history (−)	2146	10 (0.5)	18 (0.8)	26 (1.2)
COVID-19 history (+)	14	0 (0)	10 (71.4)	10 (71.4)
Age				
20–29	411	2 (0.5)	11 (2.7)	12 (2.9)
30–39	587	3 (0.5)	1 (0.2)	3 (0.5)
40–49	601	3 (0.5)	7 (1.2)	10 (1.7)
50–59	330	0 (0.0)	4 (1.2)	4 (1.2)
60–69	174	2 (1.2)	5 (2.9)	7 (4.0)
≥70	57	0 (0.0)	0 (0.0)	0 (0.0)
*p* value ^d^		0.60	0.006	0.007
Sex				
Female	1547	6 (0.4)	21 (1.4)	27 (1.8)
Male	613	4 (0.7)	7 (1.1)	9 (1.5)
*p* value ^d^		0.41	0.69	0.65
Job categories				
Nurse ^a^	651	3 (0.5)	11 (1.7)	14 (2.2)
Physician	97	0 (0.0)	1 (1.0)	1 (1.0)
Technician ^b^	369	2 (0.5)	3 (0.8)	4 (1.1)
Nursing care staff	329	2 (0.6)	8 (2.4)	9 (2.7)
Office worker	270	2 (0.7)	2 (0.7)	4 (1.5)
Receptionist	33	0 (0.0)	1 (3.0)	1 (3.0)
Clinical research unit worker	192	0 (0.0)	0 (0.0)	0 (0.0)
Other ^c^	219	1 (0.5)	2 (0.9)	3 (1.4)
*p* value ^d^		0.95	0.25	0.33

^a^ Nurses include assistants; ^b^ Laboratory technicians, radiology technicians, pharmacists, clinical engineers, dental hygienists, physical therapists, occupational therapists, speech therapists, and acupuncturists; ^c^ Drivers, security personnel, nursery school teachers, workers in shop, sanitary workers, nutritionists, and food service staff; ^d^ Pearson’s chi-square test.

## Data Availability

The data that support the findings of this study are available from the corresponding author, T.Y., upon reasonable request, according to the Ethical Guidelines for Medical and Health Research Involving Human Subjects, Japan.

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
