# Peer review of "SARS-CoV-2 Seroprevalence among Healthcare Workers in General Hospitals and Clinics in Japan"

_ijerph, 2021, doi:10.3390/ijerph18073786_

Round 1
Reviewer 1 Report
The study examines the seroprevalence of severe acute respiratory syndrome coronavirus-2 (SARS-CoV-2) antibodies among 2,160 healthcare workers in hospitals and clinics that are not designated to treat COVID-19 patients in Japan.
These results show a higher prevalence of SARS-CoV-2 immunoglobulin in the population studied than that observed in the general population in Japan, therefore indicating that healthcare workers should be the top priority for further social support and vaccination programs against SARS-CoV-2.
The manuscript is well written and needs minor revisions.
ABSTRACT Line 43: please add the value of the prevalence of SARS-CoV-2 immunoglobulin in the general population in Japan
Table 1: edit the format of the table, it cannot be properly read
Lines 119-120: add a statistical test to evaluate the presence/absence of significant differences in antibodies prevalence between the cited categories
If available, authors should indicate if the patients who tested positive for IgM on august and positive to IgG on September developed symptoms related to covid disease
Did the authors validate the assay on a limited cohort of patients represented by true negative (maybe including sera collected before the pandemic) and true positive patients (maybe including COVID-19 affected patients confirmed by viral nucleic acid RT–PCR test on nasopharyngeal swabs) in order to confirm the expected performance of the assay?
Reviewer 2 Report
This study surveyed the prevalence of COVID-19 among healthcare workers in Japan. I should appreciate the authors' time and patient to come up with some results. However, there are several problems that deduct from the quality of this manuscript. Below are several comments on this work.
- The innovations involved in this report are insufficient. I could not see much insights from it.
- The authors should adjust the Table 1 to show full contents.
- More analysis should be added into Results section but not just described the shallow figure results.
- Since you have surveyed the prevalence of COVID-19, some treatments should be provided for the hospitals.
- The questionnaire used in this study should be provided as supplementary.
Round 2
Reviewer 2 Report
The authors have addressed all my comments.